# Volume Changes in the Descending Aorta after Frozen Elephant Trunk and Conventional Hemi-Arch Repair after Acute Type A Aortic Dissection

**DOI:** 10.3390/diagnostics12102524

**Published:** 2022-10-18

**Authors:** Abdulhakim Ibrahim, Arash Motekallemi, Ahmed Yahia, Alexander Oberhuber, Thorsten Eierhoff, Sven Martens, Elena Marchiori, Andreas Rukosujew

**Affiliations:** 1Department of Vascular and Endovascular Surgery, University Hospital Muenster, 48149 Muenster, Germany; 2Department of Cardiothoracic Surgery, University Hospital Muenster, 48149 Muenster, Germany

**Keywords:** aortic surgery, frozen elephant trunk, hemi-arch repair, conventional aortic arch replacement

## Abstract

The aim of this study was to compare the mortality rates, re-intervention rates, and volumetric changes in aortas following surgery, in terms of the true lumen and false lumen changes, using conventional hemi-arch repair (CET) and frozen elephant trunk (FET) techniques. During the period from 2015 to 2018, 66 patients underwent surgical treatment for acute aortic dissection (Debakey type 1). Demographic and procedure-related data were evaluated. We measured volumetric change before surgical treatment, at discharge, and at 12- and 24-month time points based on computed tomography angiography. The study cohort was divided into two groups (FET vs. CET). The mean age of the patients was 56.9 ± 9.4 years in the FET group versus 63.6 ± 11 years in the CET group (*p* = 0.063). The mean follow-up time was 24 ± 6 and 25 ± 5 months for the FET and CET groups, respectively. There were no significant differences between the two groups in terms of the medical histories of the cohorts. The results showed a significant increase in true lumen volume after the FET procedure (within 24 months postoperatively; *p* = 0.005), and no significant changes in total (*p* = 0.392) or false lumen (*p* = 0.659) volumes were noted. After the CET procedure, there were significant increases in total and false lumen volumes (*p* = 0.013, *p* = 0.042), while no significant change in true lumen was observed (*p* = 0.219). The volume increase in true lumen after the FET procedure was higher compared to the CET group at all postoperative time points (at discharge, 12 months, and 24 months) without significant evidence (*p* = 0.416, *p* = 0.422, *p* = 0.268). At two years, the volume increase in false lumen was significantly higher among the CET group compared to the FET group (*p* = 0.02). The Kaplan–Meier curve analysis showed that patients who underwent the CET procedure underwent significantly more re-interventions due to false lumen expansion of the descending aorta (*p* = 0.047). Present study results indicate that the true and false lumen changes in the aorta following the FET and CET procedures were different. FET led to a significant increase in true lumen volume, while false lumen volume remained stable; however, after the CET procedure, significant false lumen enlargement was noted at mid-term follow-up time points. The re-intervention rate after CET was higher due to false lumen expansion.

## 1. Introduction

Despite remarkable advancements in surgical technique and the implementation of new surgical armamentaria, late complications after the surgical repair of aortic dissection involving the arch and descending aorta are most often due to persistent false lumen patency [1,2]. In order to reduce aneurysmal degeneration of the downstream aorta, various methods of open and endovascular surgery have been developed in recent years [3,4]. The frozen elephant trunk (FET) technique allows for single-stage repair, especially for acute aortic dissection type A. The FET procedure promotes the remodeling of the aorta by maintaining the true lumen flow and facilitating its expansion and promoting false luminal thrombosis [5]. In the case of chronic aortic dissection with a post-dissection aneurysm or thoracoabdominal aneurysm, a FET is also usually implanted. The frozen elephant trunk is believed to provide a solid landing zone for either endovascular or surgical thoracoabdominal repairs in Stage II. However, there are situations where CET implantation is chosen—namely, when the true lumen is very narrow or the true lumen is already occluded in the thoracic cavity. Even if the coeliac trunk or superior mesenteric artery is perfused from the false lumen in the abdominal area and it is unclear that re-entries are large enough, there is a risk of reduced perfusion of the abdomen after FET implantation. Currently, computed tomography (CT) is used to assess true and false lumen size after the surgical repair of a type A aortic dissection. However, the diameter measurement alone cannot adequately evaluate the volumetric progression of the dissected aortic lumina, since the measurement is based on each slice and not on the total volume [6]. A thorough understanding of true and false volume remodeling after the repair of the arch and the descending aorta is essential to understanding aortic remodeling. The aim of this study was to compare the mortality rate, intervention rate, and volume changes in the descending aorta after a frozen elephant trunk procedure using the Thoraflex™ hybrid prosthesis and the so-called conventional aortic hemi-arch replacement procedure for type A aortic dissection.

## 2. Materials and Methods

During the period from January 2015 to October 2018, a total of 109 patients underwent surgical treatment for Type A aortic dissection (AADA). In our study, only patients with the presence of acute aortic dissection (Debakey type 1) were included (n = 66).

Of these patients, 22 underwent surgical repair using the Thoraflex™ Hybrid Plexus 4 (Vascutek, Terumo Aortic, Scotland) and 44 patients underwent surgical repair using the conventional hemi-arch repair. Following the frozen elephant and conventional hemi-arch replacement procedures, the diametric and volumetric changes in the descending aorta recorded at the early and mid-term follow-up time points were analyzed. Surgery was performed within the Department of Cardiothoracic Surgery at Münster University Hospital. Secondary vascular procedures were performed within the Department of Vascular and Endovascular surgery at the same hospital.

### 2.1. Surgical Replacement

The replacement of the aortic arch was performed under moderate hypothermic (28 °C, on average) circulatory arrest and bilateral selective antegrade cerebral perfusion (SACP) with the application of near-infrared spectroscopy (NIRS). After the median sternotomy, extracorporeal circulation was initiated. Myocardial protection was achieved using retrograde cold blood cardioplegia. After reaching the desired temperature and circulatory arrest, either a hemi-arch repair or FET was applied. In patients with FET, the aortic arch, in zone 2 or 3, was transected distally and the endoprosthesis was deployed into the descending aorta. The size of the stent graft was chosen according to the maximum diameter of the true lumen. After the completion of the distal anastomosis, the vascular reconstruction of the aortic arch was performed with the reinitialized perfusion of the lower body [6].

### 2.2. Volume Measurement

Volume measurements were based on ECG-gated computed tomography angiograms (CTAs) taken before surgical treatment, at discharge, and at the one- and two-year follow-ups. In total, 306 volumetric measures were obtained from the left subclavian artery to the celiac trunk (true and false lumen). Acute aortic dissection (Debakey type 1) was diagnosed in all patients. For every case, we performed a volumetric reconstruction of the true and false aortic volume levels at the different time points. In the case of missing or inconsistent data, scans were not included. The true lumen volume (TL) and false lumen volume (FL) values were analyzed. First, volumetric analysis software (Aquarius iNtuition, TeraRecon, Inc., Foster City, CA, USA) was used to perform 3-D reconstructions with the volumetrics of the CT scans. The CT images were loaded into a standard multiplanar reformatting package, which then showed images in three orthogonal planes. Second, after curved planar reformation (CPR), a centerline was generated. Then, after the selection of the preset function for measuring volume, the true and false lumen volumes were divided automatically from the software by detecting the level of contrast medium enhancement. Third, the diameters of the true and false lumen were edited manually from the left subclavian artery to the celiac trunk on each slice. The true lumen and false lumen were calculated separately for each slice; FL thrombus was included. All volumes are expressed in cm^3^. Based on the previous steps, changes in true lumen volume and false lumen volume were analyzed at the respective time points. The previous volume measurement methods are shown in Figure 1 [6].

### 2.3. Statistical Analysis

All statistical analyses were performed using SPSS statistical software for Windows (version 26.0; IBM Corp., Armonk, NY, USA). The variables were investigated using analytical methods (Kolmogorov–Smirnov/Shapiro–Wilk tests) to determine the normality of their distributions. Continuous variables are expressed as means ± standard deviations (SDs) for parametric data and medians with interquartile ranges (IQRs) for non-parametric data, whereas categorical variables are presented as crude numbers and percentages. Continuous variables were compared using Student’s *t*-tests. Mann–Whitney U tests were performed for the non-normally distributed variables. Kruskal–Wallis tests were performed as nonparametric tests to test for significant differences among continuous dependent variables with categorical independent variables with two or more groups. Chi-square tests or Fisher’s exact tests were used for categorical variables. Kaplan–Meier analyses were conducted to demonstrate freedom from mortality and freedom from secondary interventions. The differences between groups were compared using Mantel–Cox log-rank tests. A *p*-value < 0.05 was considered statistically significant. 

## 3. Results

The mean age of the patients was 56.9 ± 9.4 years in the FET group versus 63.6 ± 11 years in the conventional hemi-arch repair group (*p* = 0.063). The male gender was predominant in both groups. All patients in both groups were operated on because of acute aortic dissection (Debakey type 1). The mean follow-up time was 24 ± 6 and 25 ± 5 months for the FET and hemi-arch repair groups, respectively. There were no significant differences between the two groups in terms of the medical histories of the cohorts (Table 1).

In total, 306 volumetric measures were obtained from the left subclavian artery to the celiac trunk (true and false lumen). Overall, 57 patients’ CTAs were evaluated at the time of discharge, 27 patients (40.9%) were available for re-evaluation CTAs at the one-year follow-up, and 21 (31.8%) patients were available at the two-year follow-up. 

The median volume of the true lumen in the FET group grew from 58.6 cm^3^ (29.5–732) before surgery to 133 cm^3^ (93.4–168) at the two-year follow-up. This finding shows a significant increase in the volume of the true lumen (*p* = 0.005), while the total and false lumen did not show any significant change. After hemi-arch repair, the median volume of the false lumen grew significantly from 147.5 cm^3^ (37–300) before surgery to 190.5 cm^3^ (95.5–495) at the two-year follow-up (*p* = 0.042). The total lumen volume also significantly increased from 214 cm^3^ (138–560) before surgery to 287 cm^3^ (170–560) at the two-year follow-up (*p* = 0.013). However, there was no significant increase in true lumen volume after hemi-arch repair after two years (*p* = 0.219) (Figure 2).

The volumetric measurements of the true and false lumens after FET implantation were compared with the hemi-arch repair group at different time points (Table 2). The volume of the true lumen after the FET procedure was higher at all postoperative time points (at discharge, 12 months, and 24 months) compared with the hemi-arch repair group without significant evidence (*p* = 0.416, *p* = 0.422, *p* = 0.268). At two years, the volume of the false lumen was significantly higher after the hemi-arch repair [190.5 cm^3^ (95.5–495)] compared to the FET group [133 cm^3^ (93.4–168)] (*p* = 0.020) (Table 2). 

Of the 66 patients, 4 died within 30 days, resulting in a surgical mortality rate of 6.06%. At follow-up, one further death was registered. We evaluated our patients according to Kaplan–Meier survival/re-intervention estimate curves. Mortality did not differ significantly between the groups (Figure 3A). A total of 3 patients in the FET group and 11 patients in the hemi-arch repair group received a secondary aortic procedure. The rate of re-intervention also did not differ between the two groups (Figure 3B). Only one patient in the FET group and seven patients in the hemi-arch repair group underwent surgery due to false lumen expansion of the descending aorta. Using Kaplan–Meier estimate curves, a comparison of FET and hemi-arch repair groups, according to FL-related re-intervention, showed a significant difference between the groups (*p* = 0.047) (Figure 3C).

## 4. Discussion

This study demonstrated volumetric changes, in terms of TL growth and FL regression, using conventional hemi-arch and hybrid aortic arch repair techniques. Therefore, understanding the dynamics behind the remodeling of untreated aortic segments is mandatory. The development of the FET is claimed to be the greatest improvement in the treatment of complex aortic surgery [7]. The FET procedure combines the idea of open and endovascular treatment and allows for the open reconstruction of the aortic arch and the antegrade implantation of a stent graft into the descending aorta. FET, as well as hemi-arch replacement, has advantages and disadvantages, and the careful selection of the appropriate technique should always be guided by etiology and anatomical indications [8,9]. Previous studies have outlined the diverging effects on aortic remodeling after FET and hemi-arch replacement in patients with AADA [4,10]. Persistent communication of false and true lumen, initial aortic diameter as well as partial thrombosis, among others, can influence unfavorable remodeling [11]. Our study confirms positive effects on aortic remodeling after FET in AADA patients by demonstrating a significant increase in true lumen volume while false lumen remains stable. The largest increase in the true lumen in this group has been measured between discharge and the 12-month follow-up but keeps significantly growing over time. Moreover, FET application leads to the stabilization of the thoracic aorta over a large distance and reduces re-interventions. Through antegrade implantation of the stent graft, potential intimal tears in the proximal part of the descending aorta are closed, directing blood flow into the true lumen (TL) while initiating thrombosis of the false lumen (FL) along the stent graft [12,13].

False lumen progression has been discussed to be a crucial risk factor for aneurysmal degeneration and delayed or failed remodeling. Moreover, it is associated with 25% less event-free survival [14]. In many cases, non-resection, with or without a secondary entry in the thoracoabdominal aorta, leads to FL patency. Although initial surgery aims to eliminate all potential re-entry sites and induce FL thrombosis, this might not be achieved in patients with (undetected) distal aortic re-entries undergoing hemi-arch procedures. 

The usual surgical approach for complex tears in the aorta with (several) re-entries consists of FET as the stent graft in the descending thoracic aorta and may open the true lumen, obliterate secondary entry tears, and induce FL thrombosis, resulting in a better remodeling [15,16]. In the existing literature, several studies investigated the utility of the FET procedure in promoting FL thrombosis and aortic remodeling [15,16,17]. However, volumetric changes of the thoraco-abdominal aorta after the surgical treatment of the aortic arch are not yet part of the surveillance routine, and the majority of data evaluating lumen modifications refer to two-dimensional surface records at different aortic levels in CTA- based studies. Thin-layered ECG-gated CTAs, as performed during the routine follow-up, can be used for aortic volume reconstructions, quantitative TL and FL assessments, and lumen development analyses to create a reproducible background of patient data.

We described the mid-term volume outcomes of 66 patients after type A aortic dissection. Using CTA reconstruction software, we evaluated the volume of the aortic TL and FL before surgery, at discharge, and 12 and 24 months following surgery. The median TL volume was significantly increased at discharge and 12 and 24 months following FET, whereas no significant difference was detected in the hemi-arch group. This is probably due to longer coverage of the true lumen with closed intimal tears in the proximal part of the descending aorta in the FET group. The FL and TL volume was significantly increased at discharge, and at 12 and 24 months after hemi-arch replacement, whereas no significant difference was detected in the FET group. 

Thus, a significant volumetric increase in the true lumen could be detected only in FET patients. In addition, FET patients showed no significant change regarding the volume of the false lumen, suggesting that the hybrid procedure could induce a beneficial remodeling of the aorta. It has already been described in the literature that the procedure for the replacement of the aortic arch, plus endoprosthesis in the descending aorta, does not alter the cardiac dynamics—in particular, inducing a left ventricle overload for increased resistance in a rigid aorta [18].

This evolution could have remained undetected longer. With standard measurements based on diameter and surface measurements based on center-lined aortic reconstructions, these can accurately and reproducibly be performed during follow-up CTAs using commercially existing aortic reconstruction software. Further, and most importantly, they are also retrospectively feasible, thus expanding the number of patients who could benefit from their evaluation. These assessments do not require any additional radiation, because they are based on the software-based post-processing of routine follow-up CTA imaging. Regardless of the chosen prosthesis type, careful planning of follow-up imaging and consecutive procedures should be based on an exact understanding of aortic lumen remodeling rather than rigid schedules, as there is no consensus within the current guidelines regarding the volumetric changes of the aortic lumen. As the complete process of aortic remodeling remains multi-factorial and incompletely understood, quantifying volumetric changes in the aorta following different surgical approaches might contribute to further clarification with regard to more adequate follow-up timing and subsequent procedure planning. More importantly, our data confirm that the surgical approach does significantly influence aortic remodeling, and medium- and long-term effects of aortic remodeling should be considered. 

### Limitations

Among the limitations of the current study, its single-center, retrospective characteristics as well as its small sample size should be mentioned. Due to the retrospective study design, a standardized protocol is missing, which led to the loss of patients with CTAs at the mid-term follow-up. This also may have biased our data from follow-up CTAs for the one- and two-year follow-up time points.

## 5. Conclusions

The observed volumetric changes suggest different trends in aortic remodeling between the FET and hemi-arch groups, with the FET group developing significantly increased TL and stable FL and the hemi-arch repair group manifesting significant FL enlargement during follow-up at 24 months. In conclusion, different follow-up protocols could be advantageous; in this context, volumetric assessment can assist in the identification of patients with a higher risk for aneurysmatic development, resulting in tighter follow-up and the prevention of aortic complications.

## Figures and Tables

**Figure 1 diagnostics-12-02524-f001:**
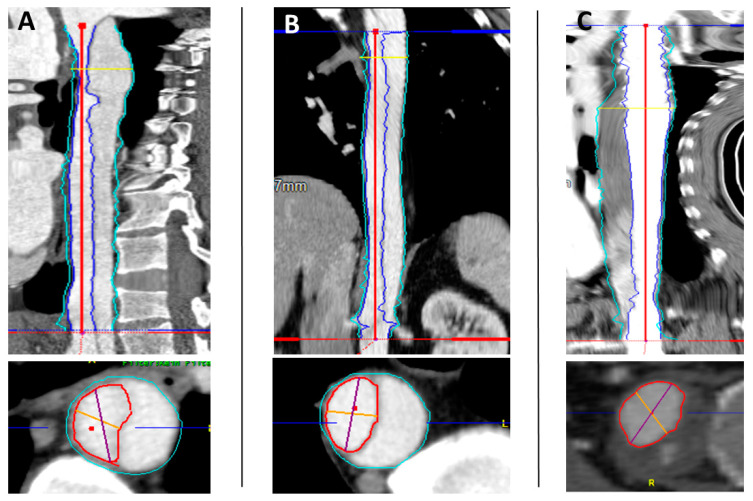
Exemplary CT displaying curved planar and straight multiplanar reconstruction before surgery (**A**), 24 months after hemi-arch repair (**B**), and 24 months after FET implantation (**C**) according to the aforementioned stepwise approach using Aquarius iNtuition (TeraRecon Inc., Foster City, CA, USA). Red line: edited centerline. True lumen was selectively marked and measured separately and manually for each CT slice before and after hemi-arch repair/FET implantation (marked red area). Blue line marks the true lumen in the upper part of the figures. Turquoise line marks the total lumen.

**Figure 2 diagnostics-12-02524-f002:**
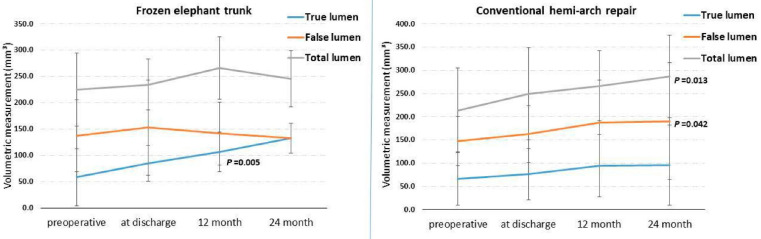
Median lumen in cm^3^ at the respective time of measurement.

**Figure 3 diagnostics-12-02524-f003:**
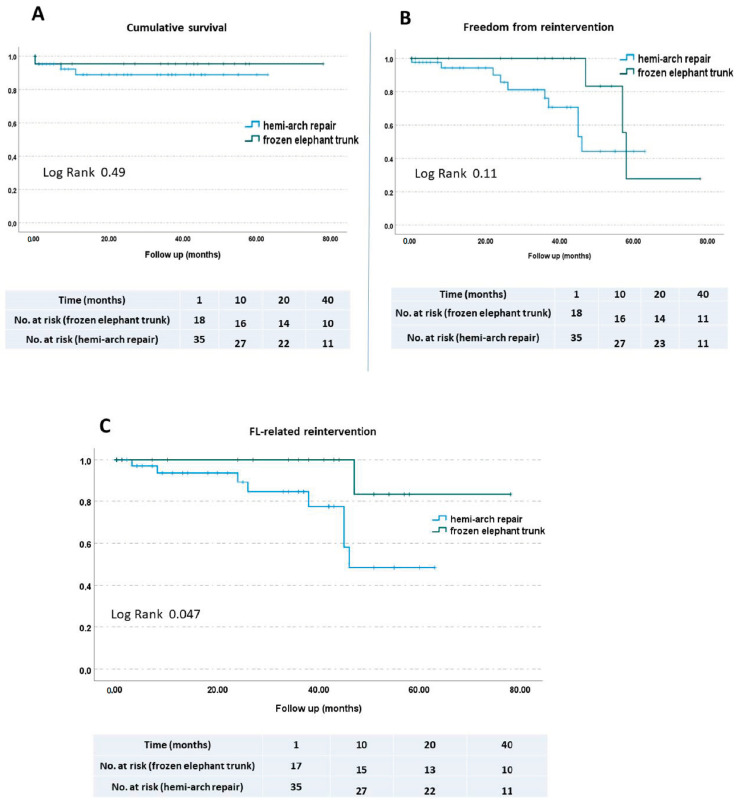
Kaplan–Meier survival and re-intervention estimates showed no significant differences between the groups (**A**,**B**). (**C**) Patients in the hemi-arch repair group have a significant difference for FL-related re-intervention.

**Table 1 diagnostics-12-02524-t001:** Baseline characteristics of the study population.

	Overall (66)	FET (n = 22)	CAR (n = 44)	Significance
Demographic characteristics				
Age (years), mean ± SD	61.25 ± 10.79	56.95 ± 9.4	63.65 ± 11	0.063
Female gender, n (%)	18 (7.2)	2 (9)	16 (36.3)	0.038
Body mass index (kg/m^2^),	26.45 ± 5.36	25.5 ± 3.95	28 ± 5.9	0.287
Medical history, n (%)				
Hypertension	50 (75.7)	15 (68.1)	35 (79.5)	0.367
Previous stroke/TIA	7 (10.6)	3 (13.6)	4 (9)	0.678
COPD	5 (7.5)	1 (4.5)	4 (9)	0.658
NYHA III or NYHA IV	7 (10.6)	0 (0)	7 (15.9)	0.086
Current/previous smoker	21 (31.8)	6 (27.2)	15 (34)	0.078
Atrial fibrillation	10 (15.1)	2 (9)	8 (18.1)	0.476
previous cardiac surgery	4 (6)	0 (0)	4 (9)	0.293

COPD, chronic obstructive pulmonary disease; TIA, transient ischemic attack; SD, standard deviation.

**Table 2 diagnostics-12-02524-t002:** TL, FL, and total lumen in cm^3^ at the respective times of measurement.

	Overall	FET	CET	Significance
True lumen in cm^3^, median (IQR)				
Preoperative	63.8 (29–732)	58.5 (29.5–732)	66.5 (29–226)	0.908
At discharge	78.1 (30–268)	85 (39.9–181)	76 (30–268)	0.416
12 months	102 (27–283)	107 (51–179)	94.9 (27–283)	0.422
24 months	100 (27–319)	133 (93.4–168)	96 (27–319)	0.268
False lumen in cm^3^, median (IQR)				
Preoperative	144.5 (37–336)	137 (53–336)	147 (37–300)	0.772
At discharge	158 (34–459)	153 (34–459)	162 (43–326)	0.522
12 months	169 (42–392)	142 (49–261)	187 (42–392)	0.222
24 months	162 (93.4–495)	133 (93.4–168)	190 (95.5–495)	0.020

All data are cm^3^. TL, true lumen; FL, false lumen.

## Data Availability

Not applicable.

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
