# Peer review of "Volume Changes in the Descending Aorta after Frozen Elephant Trunk and Conventional Hemi-Arch Repair after Acute Type A Aortic Dissection"

_diagnostics, 2022, doi:10.3390/diagnostics12102524_

Round 1

Reviewer 1 Report

I have read with great attention and interest the article entitled "Volume changes in the descending aorta after frozen elephant trunk and conventional hemiarchial repair after acute type A aortic dissection"

I believe that the work done by the authors is certainly of a remarkable scientific level, current and interesting.

However, I have some concerns about the paper in its present form

- Abstract: It is very uninformative, unclear, too narrative and unstructured; data regarding volumes are totally missing, too many acronyms are used without being explained

- Main text: I would like the Authors to better describe the (automatic) volume measurement system through the TeraRecon software; I would also suggest to the authors to describe more in the text the volumes (and diameters) found in the different study groups at different time intervals and not to limit themselves to reporting these data only in graphs and tables.

Author Response

Dear reviewer,

Thank you for your feedback and comments. On behalf of all the authors I enclose our replies to the various comments.

Best regards

Abdulhakim Ibrahim

Department of Vascular and Endovascular Surgery

University Hospital Münster

Albert-Schweitzer-Campus 1
48149 Münster

Germany

Tel:+49-251-8345788
E-Mail [email protected]

Reviewer 2 Report

It would appropriate to add a short text about the main target related to the choice of two methods and the problems related to the persistance / increase of the False lumen.

Author Response

(The authors gave the same response as above.)

Reviewer 3 Report

Your research is interesting. You could add that the procedure of replacement of aortic arch plus endoprosthesis in the descending aorta does not alter the cardiac dynamics, in particular inducing a left ventricle overload for increased resistance in a rigid aorta.  See: Manenti A. et al. Ann Thorac Surg 2022; 114: 1098. 

Author Response

(The authors gave the same response as above.)
